# Unforeseen crystal forms of the natural osmolyte floridoside

Andrew J. Maneffa[1], Adrian C. Whitwood [2], A. Steve Whitehouse[3], Hugh Powell[3], James H. Clark[1] & Avtar S. Matharu [1✉]

Floridoside (2-α-O-D-galactopyranosyl glycerol) is a glycerol glycoside that is biosynthesised by most species of red algae and has been implicated as an intracellular regulator of various homeostatic functions. Here, we report the identification of two unforeseen crystal forms of the ubiquitous natural osmolyte floridoside including a seemingly unheralded second anhydrous conformational polymorph and the unambiguous description of an elusive monohydrated variant. By employing a variety of thermal and spectroscopic techniques, we begin to explore both their macro and molecular physicochemical properties, which are notably different to that of the previously reported polymorph. This work advances the characterisation of this important natural biomolecule which could aid in facilitating optimised utilisation across a variety of anthropocentric applications and improve comprehension of its role in-vivo as a preeminent compatible solute.

---

[1] Green Chemistry Centre of Excellence, Department of Chemistry, University of York, Heslington, York YO10 5DD, UK. [2] Department of Chemistry, University of York, Heslington, York YO10 5DD, UK. [3] Nestlé Product Technology Centre (Nestec York Ltd.), Clifton, York YO31 8FY, UK. ✉email: avtar.matharu@york.ac.uk

Floridoside (2-α-O-D-galactopyranosyl glycerol, Fig. 1) is an important, naturally occurring glycerol glycoside biosynthesised by most species of red algae (Rhodophyta)[1]. Within these organisms, it has been implicated as an intracellular regulator of various fundamental homoeostatic functions including; oxidative stress, heat shock, polysaccharide synthesis/transient carbon storage, and most notably osmotic pressure[2,3], whereby its accumulation can effectively counteract external hypertonicity and desiccation[4,5]. Thus, while concomitantly facilitating normal intracellular activity, floridoside is often considered a so-called 'compatible solute' and represents a less-detrimental method of osmoadaptation with respect to alternative approaches (accumulation of inorganic salts, metabolically disruptive solutes, e.g., urea and/or abrupt changes in cell volume)[6,7].

Unsurprisingly then, given this utility in nature coupled with the advent of improved purification and subsequent identification[8–11], there has been a recent increase in the application of floridoside across a variety of fields. Indeed, over just the preceding decade or two, a diverse array of academic literature has described its use in, for example; osteogenesis[12], neuroinflammatory supression[13], prevention of biofouling[14], anti-complementation, and as an antioxidant[15–17]. The versatility of floridoside has also stimulated significant commercial interest, as evidenced by various patent/-applications outlining its utility either solely or as part of a composite product for inter alia algal cryoprotection[18], antibiotic potentiation[19], cosmetics[20], nutraceuticals, and viral/neoplastic treatment[21,22]. It is also worth highlighting the capacity for future floridoside generation from Rhodophyta, wherein it can represent a significant constituent (≤25 dry wt.% depending on species, growth conditions etc.)[23] and for which the harvest of several genera (e.g., *Gracilaria*, *Porphyra*) already exceeds one million tons (wet basis)[24].

In recent years, there has been a growing canon of work dedicated to the study of other ubiquitous osmolytes including; trehalose, urea, trimethylamine-N-oxide, and glycerol, which has ultimately resulted in an improved understanding of their function within nature and use within exogenous anthropocentric products and processes[25–28]. Unlike such archetypes, there remains relatively little fundamental information concerning the physicochemical properties of floridoside, with, for example, only a single crystal structure (referred to herein as Form 'I', $F_I$) ever being reported to the best of our knowledge—first by Simon-Colin et al. and later Vonthron-Sénécheau et al.[10,11]. This itself is perhaps surprising given; (i) the propensity for close structural relatives, namely polyhydroxyl pyranoses such as D-allose, D-trehalose, and D-lactose to exhibit polymorphism (even when accounting for anomerism, which is not exhibited by floridoside as an acetal) and (ii) that its existence has been acknowledged for close to 100 years[29–32]. We note that previous mention of a hydrated species appears to have been made during early work by Colin and co-workers (at least according to a machine translation of the original French texts)[29,33], however, unequivocal evidence and formal characterisation of this otherwise elusive crystal has not yet been forthcoming to the best of our knowledge.

**Fig. 1 The natural osmolyte floridoside.** 2-α-O-D-galactopyranosyl glycerol.

In the present work, we report the unambiguous identification of a previously unheralded anhydrous polymorphs of floridoside (Form 'II', $F_{II}$) in addition to a crystalline stoichiometric monohydrate (Form 'h', $F_h$), which was initially obtained directly from the surface of *Palmaria palmata*. This has been achieved using a combination of thermal (differential scanning calorimetry (DSC), thermogravimetric analysis (TGA), simultaneous thermal analysis (STA)) and spectroscopic (single crystal XRD (s-XRD), powder XRD (p-XRD), FTIR) techniques. Given the increasing industrial and academic interest in floridoside as an emerging active component and nutraceutical, there is a prescient need for an improved characterisation and comprehension of the physicochemical properties of this highly promising biomolecule. The identification and molecular level structural characterisation concerning the newly characterised crystals that are presented herein may offer new possibilities in terms of more optimal design and utilisation of floridoside within various products/processes. They also provide a deeper insight into its role as a ubiquitous compatible solute which has been implicated as a key for the in vivo regulation of osmotic pressure and desiccation within red macroalgae.

## Results and discussion

**Initial identification of new polymorphic and hydrated crystal forms.** Upon receiving the biomass, initial attention was drawn by a crystalline material that was easily observable on the untreated (other than initial dehydration after collection) macroalgal surface (referred to herein as untreated crystalline material, UCM, Fig. 2a, b). Remarkably, the $^1$H nuclear magnetic resonance (NMR) spectrum of the UCM in $D_2O$ (Supplementary Fig. 1a, b) was virtually identical to that of pure floridoside reported in the literature, giving a characteristic doublet at *ca.* 5.09 ppm[5]. This can be attributed to the equatorial anomeric proton on account of its small coupling constant (3.7 Hz), which in turn is owing to the gauche relationship between it and the axial proton on the vicinal carbon as outlined by Karplus[34]. Further investigation by STA ($N_2$, 5 K min$^{-1}$, Fig. 2c) identified a single concerted endothermic mass loss event (of 5.9 wt.%) at ~60–95 °C, indicating the vaporisation of a fugacious species (assumed to be water), which was then followed by a second, sharper endotherm at *ca.* 140 °C before thermal decomposition of the remaining organic material beginning at ~205 °C.

Separate application of higher temperature TGA (air, 5 K min$^{-1}$, Fig. 2d) also highlighted the presence of a comparatively thermally stable species in the UCM which was not driven off until ~750 °C. The increase in mass over 100 wt.% is considered to be an experimental artefact related to buoyancy, as evidenced by the result of a blank run (Supplementary Fig. 2), which showed a similar pattern. Given that the second thermal event occurred well above the expected combustion temperature range of the organic content (note the *ca.* 9 wt.% difference at 625 °C in $N_2$/air) and the lack of signal detected via NMR, it was concluded that it corresponded to the decomposition of a solid inorganic substance devoid of protons. It should be noted that minor amounts of other inorganics were also inferred given the small mass that remained following heating to 1300 °C (~1.5 wt.% when accounting for buoyancy). Using this information and; (i) assuming the volatile species was water and (ii) attributing the mass loss prior to 750 °C (*ca.* 92.5 wt.%) to either floridoside or water, a simple mass balance indicated a close to 1:1 stoichiometry of the two components (6.1 wt.% calculated).

Thermal treatment of UCM under vacuum (80 °C, ~20 mbar, 3 h) afforded a material with a similar appearance (UCM-T) and identical $^1$H NMR spectrum (Supplementary Fig. 1c, d) but which when subjected to same investigative techniques, yielded somewhat disparate results. No initial mass loss could be observed

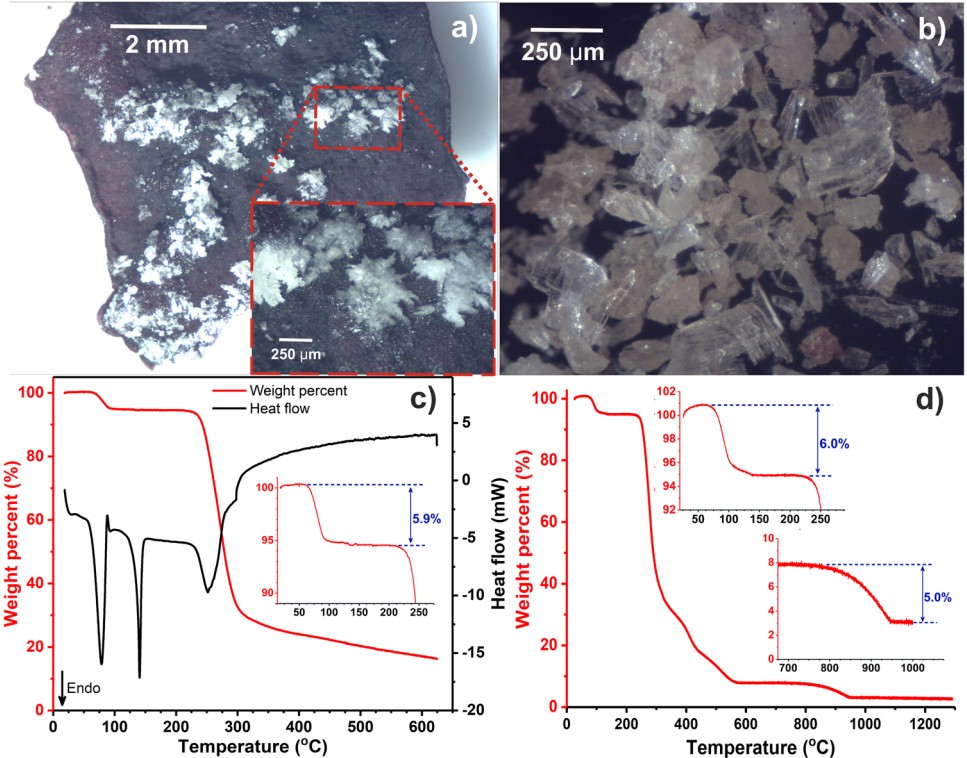

**Fig. 2 Microscopic and thermogravimetric characterisation of untreated crystalline material. a** On the surface of the as-received *Palmaria palmata* surface, **b** following mechanical removal. Traces of the corresponding simultaneous thermal analysis (10.0 mg) **c** and thermogravimetric analysis (51.2 mg) **d**.

when UCM-T was subjected to STA however, the second endotherm was still present (albeit at *ca.* 136 °C) as was the decomposition from 205 °C onwards (Fig. 3a). The initial mass loss was also absent during high-temperature TGA although both decompositions remained consistent (Fig. 3b), suggesting that heating in vacuo had removed the water but otherwise unaffected the carbohydrate and inorganic components.

In addition to the aforementioned thermal analyses, p-XRD was also conducted on UCM and UCM-T to allow comparison with the simulated powder trace belonging to the only previously reported crystal form, $F_I$ (based on the structure deposited by Vonthron-Sénécheau et al.)[11]. Remarkably, both UCM and UCM-T exhibited unique powder patterns (Fig. 3c), which were clearly different to that simulated for $F_I$ (e.g., the absence of the strong signals at $2\Theta = 7.4/19.6°$) and proof that they were primarily constituted of two unreported forms of crystalline floridoside; a hydrated variant ($F_h$) in UCM and an anhydrous form ($F_{II}$) in UMC-T. p-XRD also indicated the presence of KCl and possibly also NaCl, as can be seen by the consistent peaks at *ca.* 28.5/40.4/50.1° (KCl) and 31.5° (NaCl), respectively. This is in-keeping with prior reports that both species (particularly, the former) have been found to be the most abundant inorganic constituents of *P. palmata*[23].

DSC studies on both UCM and UCM-T (Fig. 4a, b) illustrated further differences between the two materials. In the case of the former, the first of the two endotherms occurred at a similar temperature to that recorded using STA, which was followed immediately by an apparent exothermic deviation from the baseline at *ca.* 87.5 °C and then a second, significantly broader endotherm from 93 to 118 °C. During the second heating cycle, there is a clearly observable glass transition at −12.4 °C, signifying the presence of an amorphous melt (which then appeared to undergo partial cold crystallisation and subsequent re-melting upon heating). Conversely, UCM-T exhibited only a single broad

endotherm at *ca.* 115–142 °C and a well-defined glass transition ($T_g$) of 30.1 °C, highlighting the considerable plasticisation by water which is comparable to that reported for other molten carbohydrate matrices[35].

In a further experiment, UCM was heated to a temperature of 90 °C (i.e., past the initial endotherm) within a sealed DSC pan prior to cooling (UCM-90, Fig. 4c) before being removed and subjected to STA (Fig. 4d). In this case, it was difficult to observe any obvious glass transition (at least one above −80 °C) upon cooling, indicating a comparative lack of amorphous matter. Subsequent STA of UCM-90 highlighted the presence of residual water which unlike UCM, was lost gradually and over a much broader temperature range starting at *ca.* 30 °C. In addition to this, the only notable endotherm (aside from the omnipresent decomposition signal at >205 °C) in the spectrum was similar (albeit broader in part due to residual water vapourisation) to that detected in UCM-T, indicating the presence of $F_{II}$. This supposition was corroborated by attenuated total reflection Fourier-transform infrared (ATR-FTIR) analyses (Fig. 5) of UCM, UCM-T, and UCM-90 wherein it can be seen that the spectrum of thermally untreated UCM is markedly different to that of both heated samples, which instead are essentially identical.

Interestingly, the only discrepancy between the spectra of UCM-T and UCM-90 is the presence of a somewhat broad but weak vibrational band at $\tilde{\nu} \approx 1645$ cm$^{-1}$, which is attributed to the deformation modes of $H_2O$, consistent with residual water. The same absorption band is also observable in UCM, wherein it manifests at a higher wavenumber (1672 cm$^{-1}$) and takes a considerably sharper form, indicating a clear difference in the water environment. This value is in-keeping with those of water deformation in similar carbohydrate crystals such as trehalose dihydrate (1678/1639 cm$^{-1}$) and maltose monohydrate (1639 cm$^{-1}$)[36,37]. This is further indication that the series of DSC signals at <93 °C in UCM correspond to the

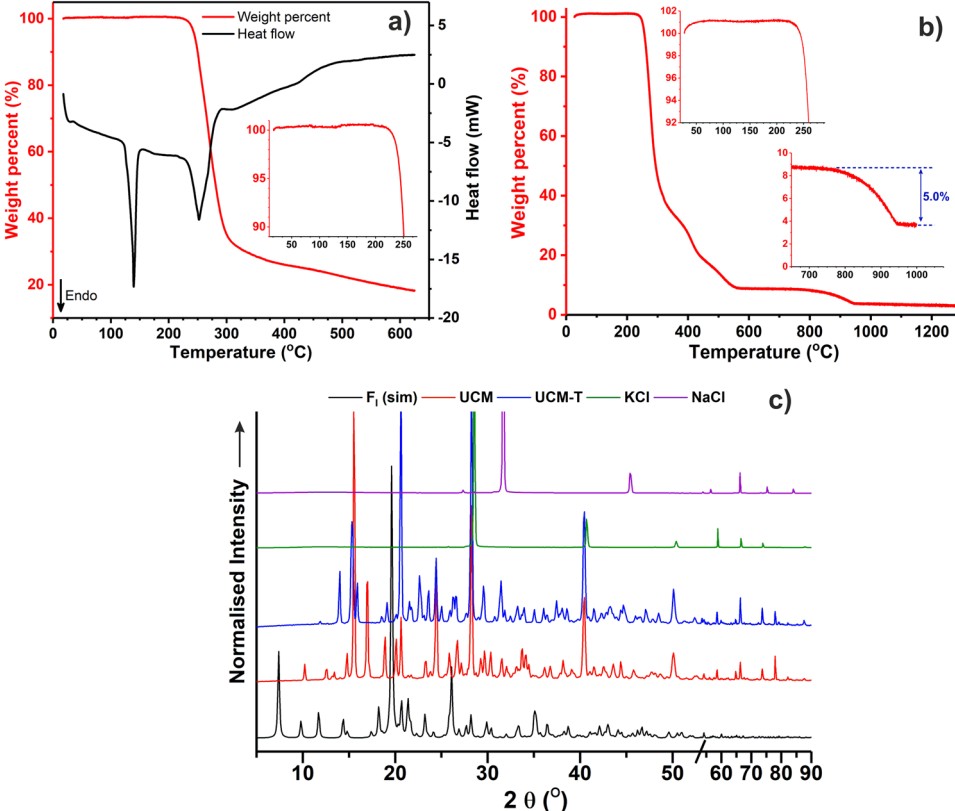

**Fig. 3 Thermal analyses of dried untreated crystalline material and comparative p-XRD. a** Simultaneous thermal analysis (10.6 mg), **b** thermogravimetric analysis (50.7 mg) of untreated crystalline material dried in vacuo, **c** experimental p-XRD traces of selected samples compared to F$_I$ (simulated using mercury where full width at half-maximum is 0.20).

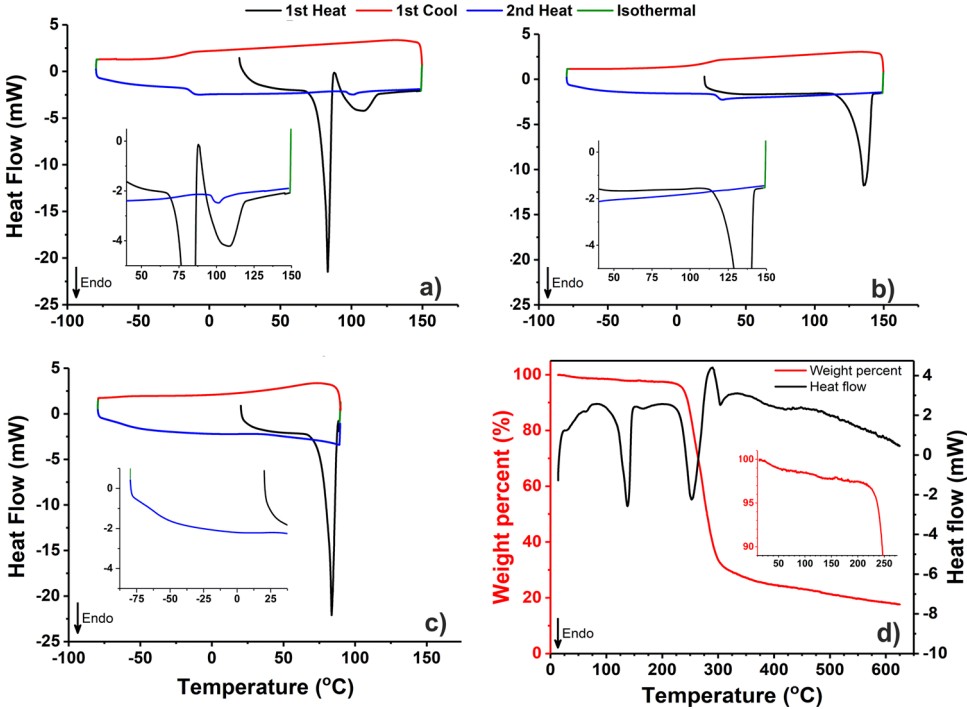

**Fig. 4 Differential scanning calorimetry of *Palmaria palmata*-derived materials.** First heating, first cooling and second heating cycles of; **a** untreated crystalline material (10.7 mg), **b** untreated crystalline material dried in vacuo (10.0 mg), **c** untreated crystalline material heated to 90 °C before cooling (11.8 mg), **d** simultaneous thermal analysis of the latter (11.0 mg).

deformation of $F_h$ and eventual formation of crystalline $F_{II}$ (then liquified upon further heating *cf.* broad endotherm) although the exact sequence and nature of the thermal events is not yet fully elucidated and should be investigated further.

**Physicochemical characterisation and future outlook**. On account of the high purity of UCM, it was possible to generate all of the floridoside polymorphs using basic heating and solvent manipulations. The typical approach consisted of boiling UCM in MeOH (initial concentrations of *ca.* 25 mg mL$^{-1}$) and hot filtration of the resulting suspensions followed by either storage or further treatment of the filtrate. The most reliable method of generating $F_{II}$ was via prolonged boiling (for several minutes) of UCM before hot filtration and natural cooling in sealed vials for 1–7 days. This generally led to clear, colourless crystals, which either grew as well-formed cuboid-like (Fig. 6a) or higher (Fig. 6b) polyhedrons, presumably depending on the magnitude of the driving force toward crystallisation.

Conversely, immediate addition of an excess of EtOAc (as an antisolvent) to the methanolic filtrate resulted in the formation of $F_I$, either concomitantly with $F_{II}$ or as the sole crystalline form depending on the relative quantities of the EtOAc and filtrate. In

the former case, diffractable needle-like crystals (Fig. 6c) could only be prepared via addition of relatively smaller amounts of EtOAc (~3:1, v/v), which resulted in the formation of clear solutions following antisolvent addition and from which, the $F_I/F_{II}$ crystals could then be manually separated. Increased addition of EtOAc so as to induce turbidity (~5:1, v/v) reliably led to the formation of spherulite-type clusters containing very fine needles (Fig. 6d), which were collected quickly following natural settling of the suspension (2–3 h) in order to minimise any potential solvent-mediated polymorphic transformation to $F_{II}$. It should be noted that further increasing the amount of EtOAc resulted in the liberation of a globular precipitate from solution, which likely corresponded to amorphous matter, although this was not investigated further.

In the case of $F_h$, large colourless crystals (Fig. 6e) could be obtained via partial slow evaporation (over several days) of the MeOH filtrate under a fume hood at room temperature. However, this approach was somewhat aleatoric and occasionally resulted in the generation of anhydrous forms or a liquid that was devoid of crystals. This is indicative of a delicate interplay between the relative concentrations (viz. thermodynamic activities) of the filtrate constituents (namely, floridoside, $H_2O$, MeOH, and salts) as has been reported for the crystallisation of other water-soluble solutes from alcohol/water solutions[38].

The newly identified crystal structures of $F_{II}$, $F_h$ are displayed alongside $F_I$ (for which the CCDC data provided by Vonthron-Sénécheau et al. has been used) in Fig. 7 (full data is given in Supplementary Table 1)[11]. All three crystals are orthorhombic and belong to the $P2_12_12_1$ space group with each unit cell comprising four floridoside molecules in addition to four $H_2O$ molecules in the case of $F_h$ as a stoichiometric monohydrate. The unit cell axes (Supplementary Fig. 3) of both $F_{II}$ ($a = 8.54811(10)$ Å, $b = 9.19251(10)$ Å, $c = 14.34851(17)$ Å) and $F_h$ ($a = 8.22038$ (16) Å, $b = 11.2533(3)$ Å, $c = 12.9852(2)$ Å) display relatively comparable lengths when compared with $F_I$, wherein one axis is markedly longer and another noticeably short ($a = 4.88440(10)$ Å, $b = 9.7259(3)$ Å, $c = 23.8754(6)$ Å) on account of an almost head-to-tail arrangement of molecules, which results in a very high aspect ratio. This is likely to be the origin of the consistent needle-like morphology, which was observed for all of the $F_I$ crystals.

From a molecular perspective, the pyranose ring adopts the $^4C_1$ conformation in all three forms, with each existing as *gauche-trans* rotamers with respect to the $C_5$-$C_6$ bond, displaying a $O_6$-$C_6$-$C_5$-

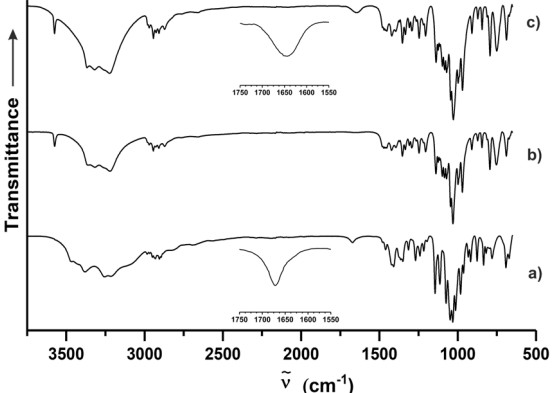

**Fig. 5 ATR-FTIR spectra of *Palmaria palmata* derived materials. a** Untreated crystalline material, **b** untreated crystalline material dried in vacuo, **c** untreated crystalline material heated to 90 °C within a sealed DSC pan. Insets magnify the 1550–1750 cm$^{-1}$ region for **a** and **c**. Arrow indicates increasing transmittance.

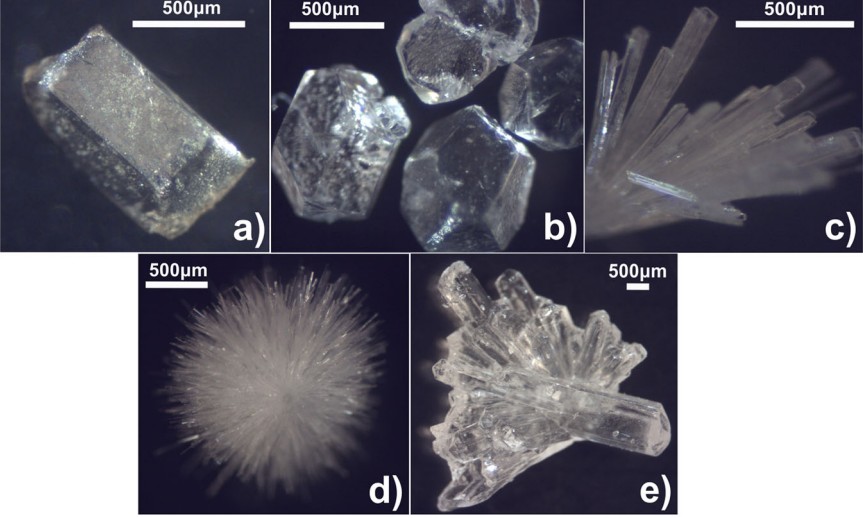

**Fig. 6 Isothermal optical micrographs of crystalline floridoside. a** and **b** $F_{II}$, **c**, and **d** $F_I$, **e** $F_h$.

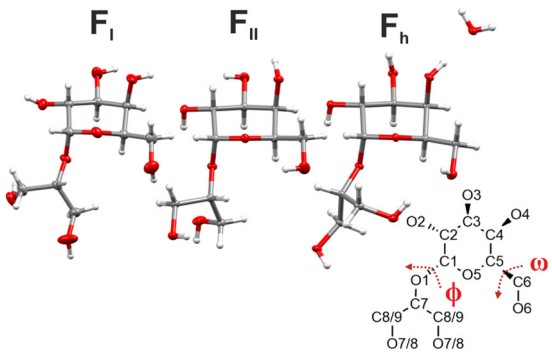

**Fig. 7 Thermal ellipsoid representations for $F_I$, $F_{II}$, and $F_h$.** Ellipsoids are shown at the 50% probability level wherein hydrogens are drawn with arbitrary radii. Torsion angles $\omega$ and $\phi$ are depicted to aid visualisation.

$O_5$ torsion ($\omega$) angle of 67.99(7), 64.49(18), and 58.27(18)° in $F_I$, $F_{II}$, and $F_h$, respectively, (structural overlays are shown in Supplementary Fig. 4). Interestingly, this conformation (as opposed to *gauche-gauche* or *trans-gauche*) appears to be common across many naturally occurring α-substituted D-galactopyranoses (e.g., methyl α-D-galactopyranoside monohydrate = 62.60°[39], α-D-galactopyranosyl-α-D-galactopyranoside = 63.83°[40], α,β-melibose monohydrate = 65.69°)[41]. Aside from the obvious differences resulting from the additional presence of water (in $F_h$), the most discernible structural discrepancies of the floridoside molecule originate from the orientation of the pyranose-based hydroxyl groups and configuration of the glycerol moieties. The latter is exemplified in the case of the glycosidic bond, whereby the torsion angle ($\phi$, $O_5$-$C_1$-$O_1$-$C_7$) of $F_h$ at 91.02(16)° is considerably higher than that of either $F_I$ (73.89(6)°) or $F_{II}$ (63.39(17)°) and sufficiently so as to enable the rotation of a terminal glycerol OH toward $O_5$/$O_6$ (Fig. 7).

Satisfyingly, p-XRD patterns simulated from the single crystal data (Supplementary Fig. 5) of $F_h$ and $F_{II}$ were very similar to the experimental traces recorded for UCM and UCM-T, again confirming their identity and surprising purity. This was also the case for the corresponding ATR-FTIR spectra (Supplementary Fig. 6), which were identical to those presented in Fig. 5 but both wholly different to the spectrum of $F_I$. In addition to these spectroscopic discrepancies, the thermal properties of each of the three polymorphs were also found to be considerably disparate—as illustrated by DSC and hot-stage optical microscopy (Fig. 8). In the case of $F_I$ (Fig. 8a), a clear melting endotherm with an extrapolated onset ($T_o$) and maximum fusion temperature ($T_m$) of 128.1 and 130.0 °C can be observed during the first heating cycle, consistent with previous reports (127–134 °C)[10,11]. This is followed by a smaller exothermic signal, which is attributed to the formation of crystalline $F_{II}$, which then proceeds to melt at *ca.* 140 °C—in good agreement with the $T_o$ (139.3 °C) and $T_m$ (142.2 °C) found for the pure $F_{II}$ crystal (Fig. 8b). In both cases, there is clear indication of glass formation upon melt quenching, with a reproducible $T_g$ of *ca.* 32 °C. The aforementioned results also seem to indicate a monotropic relationship between the $F_I$ and $F_{II}$ polymorphs, whereby the irreversible transition from the meta ($F_I$) to stable ($F_{II}$) dimorph occurs either through a solid–solid or liquid–solid process close to or above the fusion $T$ of the former. It should be noted that neither such transformation was observable during hot-stage microscopy of $F_I$ (a full sequence of micrographs from 45 to 139 °C is provided in Supplementary Fig. 7) however, which is not unexpected given the kinetic nature of these events. In both crystals systems, it was also possible to observe melting of a comparatively small quantity of $F_I$ at *ca.*

130 °C during the second heat step (see insets), which solidified from the amorphous phase during the heat/cool/heat cycle.

Interestingly, the large, well-formed $F_h$ crystals exhibited only a single large endotherm with $T_o$ and $T_m$ values of 86.0 and 89.1 °C (Fig. 8c) upon heating to 150 °C. Heating past this signal (to 100 °C) followed by immediate cooling led to the formation of a glass with a $T_g$ of ~−11.2 °C (which underwent cold crystallisation during the 2nd heat cycle—see Supplementary Fig. 8), similar to that found for molten UCM (−12.4 °C), confirming that a significant solid–liquid transition had occurred. Indeed, this was also corroborated by the observation of an isotropic melt during hot-stage optical microscopy, which subsequently persisted through further heating (the entire experimental range of 35–135 °C is shown in Supplementary Fig. 9). This thermal behaviour is notably different to that described earlier for UCM (also constituted of $F_h$), wherein crystalline $F_{II}$ was instead formed upon initial heating. It is tentatively hypothesised that this discrepancy is related to the greater structural integrity of the well-formed, single $F_h$ crystals, which ultimately results in comparatively retarded molecular dynamics of the crystalline components upon deformation, preventing expedient lattice rearrangement, which is necessary for the formation of the anhydrate. It is likely that such behaviour is highly dependent on experimental conditions, as has been reported for the thermally induced deformation of trehalose dihydrate and which seemingly necessitates the need for more detailed future studies[42].

Unsurprisingly, given the abundance of OH groups in the floridoside molecule, all three crystals exhibit a complex spatial network of hydrogen bonding (H-bonding). To probe for potential differences in the H-bonding of the three different forms, a relatively lenient (albeit arbitrary) criteria was applied, wherein the maximum distance and minimum angle between the OH donor hydrogen and acceptor ($R_1$-O-$R_2$) were limited to ≤3 Å and ≥90°, respectively. Extending the definition past these limits led to a rapid increase in the number of possible contacts, many of which appeared to be unrealistic and hence, were not considered as was also the case for C-H donors (which have been reported to exist in similar systems)[43]. Interestingly, these values have some basis in previous relevant literature with Steiner and Saenger independently concluding that it was difficult to discriminate between H-bonding and non-bonding regions outside the same cutoffs during their analysis of 15 different non-ionic carbohydrates[44]. Indeed, the geometric boundaries currently employed may actually be overly inclusive, as highlighted by the fact that the $H_2O$ molecule in $F_h$ unrealistically forms five H-bonds (Supplementary Fig. 10).

Enacting the aforementioned criteria, the H-bonds were found to fall into one of two general groups. The first would conventionally be described as 'strong', displaying lengths and angles of ≤2.20 Å and ≥140°, respectively. The second, which are comparatively 'weaker' in nature are >2.20 Å and <140° and likely to be more ambiguous with respect to their actual manifestation within the crystal. It is interesting to note that all hydroxyls in $F_I$ are involved in both strong intermolecular hydrogen bond donation and acceptance (Fig. 9d), as highlighted previously by Vonthron-Sénécheau et al.[11] but also that only a single weak intramolecular H-bond could be identified when using the same description (Fig. 9a). In contrast to $F_I$, $F_{II}$ was found to display greater and considerably more varied H-bonding, containing many bi-/trifurcated configurations (both intra/intermolecular), which may contribute to the greater stability of the latter as reflected in the higher fusion temperature. It should be noted however, that the sharp peak at *ca.* 3576 cm$^{-1}$ in the IR spectra (Fig. 5 and Supplementary Fig. 6) indicates the presence of a single, isolated hydroxyl group, which again suggests that not all

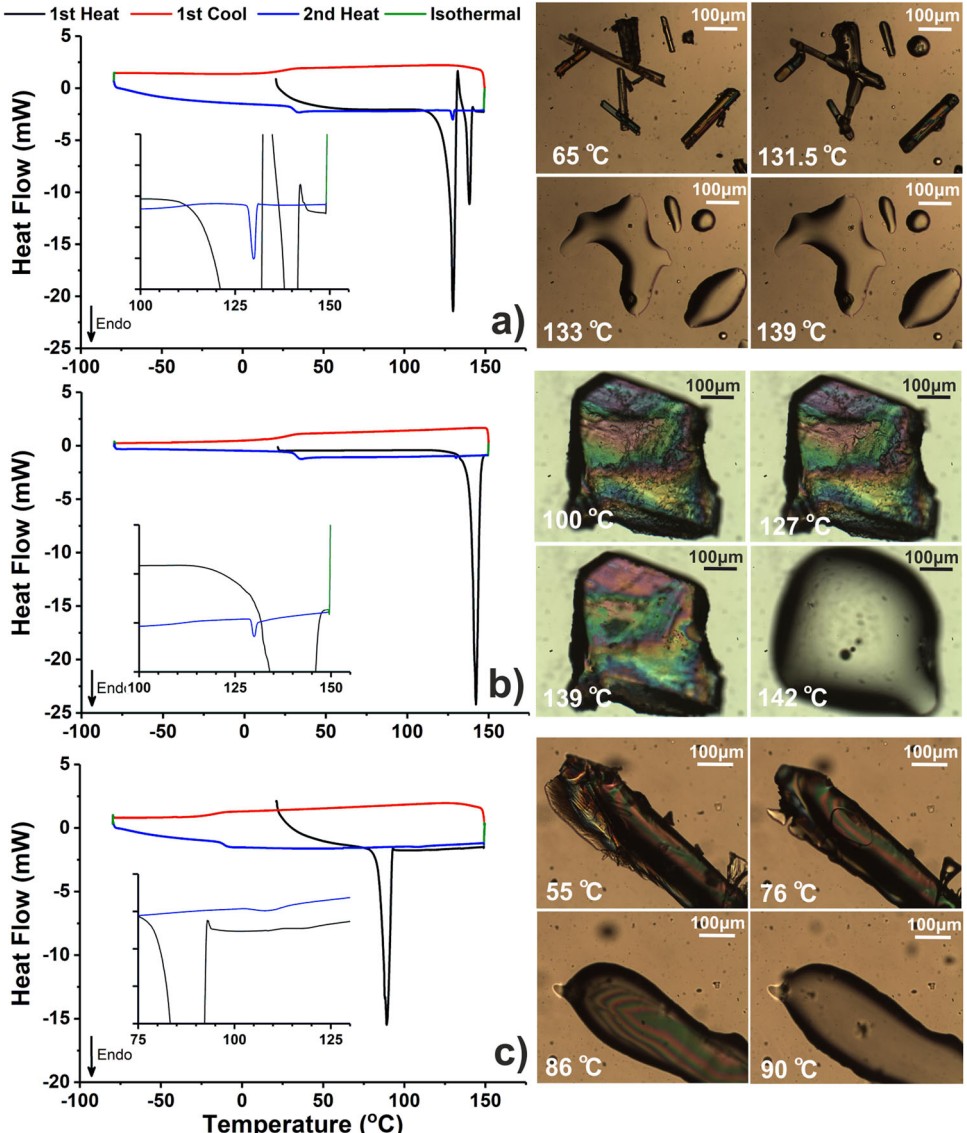

**Fig. 8 Cyclic DSC and hot-stage micrographs of floridoside crystals. a** $F_I$ (8.2 mg), **b** $F_{II}$ (8.8 mg), **c** $F_h$ (6.0 mg). Insets within the DSC traces highlight the 100–150 °C **a**, **b** or 75–130 °C **c** regions wherein y axis ticks correspond to 0.5 mW.

of the H-bonds identified in Fig. 9 actually manifest within the crystal.

In $F_h$, all but one of the OH groups are engaged in strong intermolecular donation and acceptance with either a second floridoside or water molecule, with the only exception being that one of the glycerol hydroxyls instead acts as a strong intramolecular donor to O6, around which sufficient space exists owing to the aforementioned wider glycosidic torsion angle (Fig. 9c). The same hydroxyl forms a second, weaker contact to the ring oxygen with two further instances of weak intramolecular H-bonding (O(3)H···O(4) and O(2)H···O(1)) also being detectable in $F_h$.

It is hypothesised that the formation of this hydrate may have some utility in vivo as a means of regulating the intracellular concentration and distribution of water, akin to what has been proposed for trehalose dihydrate[25]. This would provide an effective control of internal osmotic pressure, especially within specific domains wherein the local floridoside content may be particularly high, such as the cytoplasm[45,46]. In addition to this, some authors have also implicated floridoside synthesis as a mitigation strategy for desiccation stress (during low tide) in

various red macroalgae, although this may not be universally adopted across all species[4,47–49]. This again is very reminiscent of the trehalose accumulation that has been reported in other organisms (tardigrades, nematodes etc.) during periods of anhydrobiosis. It has been postulated that the functionality of trehalose could be related to the reversible un/loading of water within structured channels that exist within the dihydrate[25,31]. Inspection of the $F_h$ lattice indicates that the formation of a connective water network is unlikely given the large distance between neighbouring $H_2O$ molecules ($\geq 6.56$ Å O···O), suggesting a fundamental difference in the protective mechanism at the molecular level. This is further supported when considering that the glass transition of floridoside is far lower than that reported for trehalose (ca. 32 °C vs. ca. 100–115 °C), which makes in vivo vitrification by the former far less realistic in comparison[50].

Concerning anthropocentric use of floridoside, the identification and characterisation of metastable and hydrated crystalline species is noteworthy given that individual polymorphs and hydrates often exhibit very disparate physicochemical behaviours as exemplified by other carbohydrates such as D-glucose and D-lactose[51,52]. The metastable crystal of a dimorphic pair often

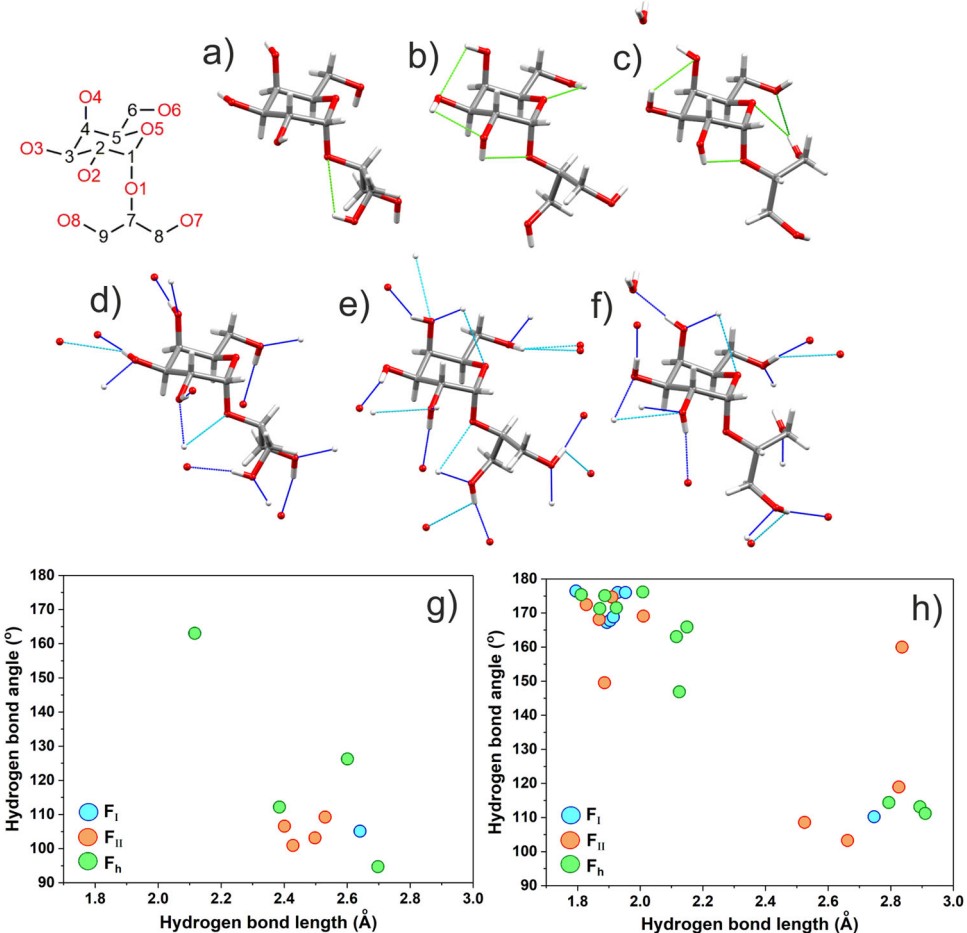

**Fig. 9 Calculated H-bonding profiles for floridoside crystals.** Showing 'strong' (<2.20 Å and >140°) and 'weak' (2.20–3.00 Å and 90–140°) intramolecular contacts (strong = dark green dashed line, weak = light green dashed line) found for **a** $F_I$, **b** $F_{II}$, and **c** $F_h$ **a**nd also intermolecular contacts (strong = navy dashed line, weak = cyan dashed line) found for **d** $F_I$, **e** $F_{II}$, and **f** $F_h$. Graphically summarised in **g** (intramolecular H-bonds) and **h** (intermolecular H-bonds). Calculations were made using Mercury wherein only OH donors/O acceptors and intramolecular contacts between atoms separated by >1 bond were permitted.

displays a comparatively faster dissolution rate but greater hygroscopicity for instance, which suggests that controlled crystal engineering towards either $F_I$, $F_{II}$, and/or $F_h$ should facilitate more optimised floridoside-containing product/processes that can be tailored towards an individual end application. Given this, more detailed studies that further explore the influence of crystalline structure on such properties will be invaluable and hence, have been initiated.

In conclusion, despite its extensiveness in nature and significant history, there has remained a lack of detailed characterisation concerning the preeminent compatible solute floridoside. Herein, we have presented unambiguous characterisation of two unforeseen crystal forms of this important biomolecule; an unheralded anhydrous polymorph and also a monohydrated variant, which has seemingly remained elusive until now. The acknowledgement of these newly identified forms of floridoside—which display distinct physical and thermal properties, is of considerable industrial and academic significance given both the increasing commercial interest and recognition of its role in vivo as a promising versatile and natural compound.

## Methods

**Materials**. *P. palmata* was foraged off Otterton Ledge, East Devon, UK (approx. Latitude: 50.6273°, Longitude: −3.2955°) in November 2018 after growing naturally in seawater (*ca*. 15 °C and salinity of 34%). Once collected, the macroalgae was partially dried using a commercial dehydrator and subsequently stored in sealed bags (up to 12 months) under ambient conditions prior to further use. Methanol (MeOH, AR grade, Fisher), ethanol (EtOH, AR grade, Fisher) ethyl acetate (EtOAc, SLR grade, Fisher), and $D_2O$ (99.9% atom D, Sigma Aldrich) were used without further purification. Sodium chloride (>99.5%, Fisher) and potassium chloride (≥99%, Sigma Aldrich) were separately dissolved in deionised $H_2O$ and reprecipitated using EtOH before being filtered, liberally washed with EtOH and dried in vacuo (5 h, 80 °C, *ca*. 20 mbar) before use.

**Preparation of floridoside-containing materials**.

(i)  UCM and UCM-T: Untreated (pale-pink coloured) crystalline material (UCM) was produced via physical scraping of the as-received biomass surface, with an aliquot thereof being subjected to drying in vacuo (80 °C, 20 mbar, 3 h) leading to the generation of a second solid (UCM-T). Both materials were stored in sealed glass vials until further use, with UCM-T being used within ≤7 days of preparation. Any visible pieces of biomass flesh that remained following collection were discarded prior to analysis.

(ii)  Crystalline floridoside Form II ($F_{II}$): UCM was boiled in neat MeOH (*ca*. 25 mg mL$^{-1}$) before hot-filtering (cotton wool plug) to a give a pale-yellow solution, which was stored in a sealed glass vial and allowed to naturally cool to room temperature (21 °C). Transparent, colourless polyhedral crystals could typically be observed following ambient storage over ~1–7 days. Crystals used for further analysis were removed from the mother liquor, collected via vacuum filtration and washed with EtOAc before being dried *in vacuo* (≥2 h, 80 °C, 20 mbar) prior to use.

(iii)  Crystalline floridoside Form h, $F_h$: The crystal submitted for XRD analysis was prepared as described for $F_{II}$ except that the MeOH filtrate was left unsealed vial and underwent partial, slow evaporation under a fume hood at *ca*. 21 °C. After a period of several days, colourless crystals could be observed

within the thickened yellow liquor. Crystals used for other analyses were removed and dried in vacuo (≥2 h, 25 °C, 20 mbar) prior to use.

(iv) Crystalline floridoside Form I, $F_I$: UCM was processed in the same way as described for $F_{II}$ except that a large excess of EtOAc (~3:1 vol:vol, room temperature) was immediately added to the MeOH filtrate. This was left overnight in a sealed vial at *ca.* 21 °C and resulted in the concomitant formation of small clusters of colourless needle-like crystals ($F_I$) and polyhedral crystals ($F_{II}$), which could be separated manually. Crystals used for all other analyses were typically spherulitic-type clusters of very fine needles prepared by immediate addition of EtOAc to the MeOH filtrate (~5:1 vol:vol) until the formation of a turbid, white suspension that was subsequently stored for 2–3 h in a sealed vial at *ca.* 21 °C. Crystals used for thermal, thermogravimetric and spectroscopic analyses were removed from the mother liquor following this time, collected and washed with EtOAc before being dried in vacuo (≥2 h, 80 °C, 20 mbar) prior to use.

**Single crystal X-ray diffraction (s-XRD)**. Diffraction data were collected at 110 K on an Oxford Diffraction SuperNova diffractometer with Cu-K$_\alpha$ radiation ($\lambda$ = 1.54184 Å) using an EOS CCD camera. The crystals (obtained directly from the crystallisation liquor) were cooled with an Oxford Instruments Cryojet. Diffractometer control, data collection, initial unit cell determination, frame integration, and unit cell refinement was carried out with 'Crysalis'[53]. Face-indexed absorption corrections were applied using spherical harmonics, implemented in SCALE3 ABSPACK scaling algorithm[54]. OLEX2 was used for overall structure solution and refinement[55]. Within OLEX2, the algorithm used for structure solution was 'SHELXT dual-space'[56]. Refinement by full-matrix least-squares used the SHELXL-97 algorithm within OLEX2[57]. All non-hydrogen atoms were refined anisotropically. C-H hydrogen atoms were placed using a 'riding model' and included in the refinement at calculated positions. O-H hydrogens were located by difference map and allowed to refine after all other atoms were located and refined. Mercury (v. 1.0) was used for the calculation of torsion angles, H-bond lengths and angles presented within the Results and Discussion section wherein H-bonds were defined as; ≤3.00 Å (H···A) and ≥90° (O-H···A), OH donor only, >1 bond separation between intramolecular donor/acceptor[58].

**Differential scanning calorimetry**. All DSC measurements were performed using a TA Instruments Q2000 mDSC operating under flowing nitrogen (50 mL min$^{-1}$) and hermetically sealed aluminium pans (TA Instruments) containing *ca.* 5–12 mg of sample referenced against an empty pan. The cyclic heat/cool method consisted of heating from 20 °C to either 90 or 150 °C before cooling to −80 °C (repeated twice further), both with 5 K min$^{-1}$ scan rate. The glass transition temperature, $T_g$ was calculated as the midpoint (to which it refers to herein) of the step-change in heat flow that occurred during the second heating cycle.

**Simultaneous thermogravimetric analysis**. STA was conducted using a PL Thermal Sciences STA 625 in which, samples (*ca.* 7–10 mg) were heated in aluminium cups under a flow of nitrogen (50 mL min$^{-1}$) from *ca.* 20 to 625 °C at 5 K min$^{-1}$.

**Thermogravimetric analysis**. High temperature TGA was performed using a Netzsch STA 409 and consisted of heating samples (~50 mg) in an alumina crucible under a flow of air (100 mL min$^{-1}$) mixed with pure N$_2$ (20 mL min$^{-1}$) at a rate of 5 K min$^{-1}$ until 1300 °C.

**Powder X-ray diffraction**. p-XRD measurements were conducted on ground material using a Bruker D8 powder diffractometer equipped with a Cu source operating under ambient conditions (power and current were 40 kV and 40 mA, respectively) over a 2Θ of 5–90° with a scanning speed of 2° min$^{-1}$ and increment of 0.1°. All simulated p-XRD spectra were generated from the s-XRD structures using Mercury (v 4.1.0) with a step of 0.1°[53].

**Attenuated total reflectance fourier transform infrared spectroscopy**. ATR-FTIR analyses were performed using a Perkin Elmer FTIR Spectrum 400 spectrometer operating in transmittance mode at a resolution of 4 cm$^{-1}$, with an acquisition of 16 scans per sample and a blank background subtraction for each experiment. Baseline corrections were performed using OPUS software.

**NMR spectroscopy**. All NMR spectroscopy was performed using a JEOL JNM-ECS400A spectrometer operating at a frequency of 400 MHz and temperature of *ca.* 25 °C. All measurements were conducted within D$_2$O (at concentrations of ~40 mg g$^{-1}$ solvent) and referenced with respect to the partially non-deuterated solvent peak (4.75 ppm) with samples being filtered through a cotton wool plug prior to analysis.

**Optical microscopy**. Hot-stage optical microscopy was performed using a Zeiss Axioskop 40Pol microscope, which was temperature controlled (to provide a constant heating a rate of 5 K min$^{-1}$ in all cases) via a Mettler FP82HT hot stage and Mettler FP90 central processor. Samples were loaded on glass slides with cover slips placed on top and observed under linearly polarised light (uncrossed polarisers) and photomicrographs were captured using an InfinityX-21 MP digital camera mounted atop the microscope. Isothermal morphological characterisation of the three floridoside crystals was conducted using an Olympus BH2 microscope operating in reflected light mode equipped with digital camera image capture (Visicam) and under ambient conditions (21 °C).

## Data availability

The X-ray crystallographic coordinates for structures reported in this Article have been deposited at the Cambridge Crystallographic Data Centre (CCDC), under deposition numbers CCDC 2004260 ($F_h$) and 2004259 ($F_{II}$) and the relevant CIFs are provided as files entitled Supplementary Data 1 ($F_h$) and Supplementary Data 2 ($F_{II}$), respectively. These data can be obtained free of charge from The Cambridge Crystallographic Data Centre via www.ccdc.cam.ac.uk/data_request/cif. All other data used within both the main manuscript and supplementary information are available from the corresponding author upon request.

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

## Acknowledgements

A.S.M. wishes to thank Nestlé and the BBSRC for provision of a CASE studentship (A.J.M.). The authors acknowledge Dr. Richard Mandle for conducting hot microscopy, Tony Coulson of Ebbtides (Devon, UK) for the provision of P. palmata, and Mr. Paul Elliott for stimulating discussions regarding thermal and thermogravimetric analyses.

## Author contributions

A.J.M. conceived of the study, performed the experimental work, analysed the data and draughted the manuscript under the supervision of A.S.M and J.H.C. Single crystal X-ray diffraction experiments, generation and refinement of the single crystal structural data were conducted by A.C.W. Isothermal optical microscopy was performed by A.S.W and A.J.M. Both A.S.W. and H.P. analysed the data and revised the manuscript. J.H.C. and A.S.M. analysed the overall data, supervised A.J.M., and revised the manuscript.

## Competing interests

The authors declare no competing interests.
