## [Peer Review File · Communications Chemistry]

REVIEWERS' COMMENTS:

Reviewer #1 (Remarks to the Author):

The present manuscript described a characterization of the natural biomolecule osmolyte floridoside. The variety of thermal and spectroscopic techniques have been employed to determine the physicochemical properties of the reported polymorphs. The manuscript is complete and well structured. The manuscript should be published in its present form.

Following minor points could be changed:

Define STA and TGA at page 5. DSC at page 8.

Include carbon names in Figure 7.

Reviewer #2 (Remarks to the Author):

The heteroside molecules , floridoside and the others related compounds (isofloridoside and digineaside) extracted mainly from red seaweeds , have many importants functions in the algal cell and particularly for the adaptation to the abiotic stresses (salinity stress for exemple)... Another in vivo function for the floridoside is as a potential source of carbon in metabolic pathway (cell wall biosynthesis).

In the last decade , natural floridoside have been investigated for the blue biotechnology applications with an interesting focus for potential medical applications (immunology and hematology.....)

The results presented by the authors are an important contribution to the khwnoledge of the physicochemical properties of floridoside. The thermal and spectroscopic techniques are well documented and added news informations on the galactosyl 2 glycerol structural forms. The publication is also important to improved the chemical synthesis of floridoside wich is also recently investigated.

Bibiography is adequat and in accordance with the revue

I recommand this work for publication in this chemical revue

The 6 of july 2020

Professor Eric Deslandes

Response to Reviewers' Comments: Unforeseen crystal forms of the natural osmolyte floridoside

Reviewer #1 (Remarks to the Author):

“The present manuscript described a characterization of the natural biomolecule osmolyte floridoside. The variety of thermal and spectroscopic techniques have been employed to determine the physicochemical properties of the reported polymorphs. The manuscript is complete and well structured. The manuscript should be published in its present form.”

Authors' Response: We thank the reviewer for reading and reviewing the manuscript. We are encouraged by the comment regarding suitability for publication.

Reviewer 1: Following minor points could be changed:

Define STA and TGA at page 5. DSC at page 8.

Include carbon names in Figure 7.

Authors' Response: We agree both points made by the reviewer and have updated the manuscript in order to incorporate these suggestions.

Reviewer #2 (Remarks to the Author):

“The heteroside molecules , floridoside and the others related compounds (isofloridoside and digineaside) extracted mainly from red seaweeds , have many important functions in the algal cell and particularly for the adaptation to the abiotic stresses (salinity stress for exemple)... Another in vivo function for the floridoside is as a potential source of carbon in metabolic pathway (cell wall biosynthesis).

In the last decade , natural floridoside have been investigated for the blue biotechnology applications with an interesting focus for potential medical applications (immunology and hematology.....)

The results presented by the authors are an important contribution to the knowledge of the physicochemical properties of floridoside. The thermal and spectroscopic techniques are well documented and added new information on the galactosyl 2 glycerol structural forms. The publication is also important to improved the chemical synthesis of floridoside wich is also recently investigated.

Bibliography is adequate and in accordance with the revue.

I recommend this work for publication in this chemical revue.”

Authors' response: We thank the reviewer for reading and reviewing the manuscript. We are encouraged by the comment regarding recommendation for publication. There were no points to address.